



# A semi-parametric hourly space-time weather generator

Ross Pidoto[1] and Uwe Haberlandt[1]

[1]Institute of Hydrology and Water Resources Management, Leibniz University Hannover, Germany

**Correspondence:** Ross Pidoto (pidoto@iww.uni-hannover.de)

**Abstract.** Long continuous time series of meteorological variables (i.e. rainfall, temperature and radiation) are required for applications such as derived flood frequency analyses. Observed time series are however generally too short, too sparse in space, or incomplete, especially at the sub-daily timestep.

Stochastic weather generators overcome this problem by generating time series of arbitrary length. This study presents a major revision to an existing space-time hourly rainfall model based on a point alternating renewal process, now coupled to a k-NN resampling model for conditioned simulation of non-rainfall climate variables.

The point based rainfall model is extended into space by the resampling of simulated rainfall events using a simulated annealing optimisation approach. Large station networks ($N > 50$) are now able to be modelled with no significant loss in the spatial dependence structure.

Modelling of non-rainfall climate variables, i.e. temperature, humidity and radiation, is achieved using a non-parametric k-nearest neighbour (k-NN) resampling approach, coupled to the space-time rainfall model via rainfall state. As input, a gridded daily observational dataset (HYRAS) was used. A final disaggregation step was then performed on all non-rainfall climate variables to achieve an hourly output temporal resolution.

The proposed weather generator was tested on 400 catchments of varying size (50 - 20,000 km$^2$) across Germany, comprising 699 sub-daily rainfall recording stations. Results indicate no major loss of model performance with increasing catchment size, and a generally good reproduction of observed climate and rainfall statistics.

## 1 Introduction

Stochastic simulation of rainfall has long been an extensive topic of research with applications including hydrological design, agricultural and water balance models, and for hydrological modelling for derived flood frequency analysis. Through regionalisation techniques, stochastic rainfall models may also be used to generate synthetic time series for ungauged sites.

At the daily timestep, a common approach is to model first the rainfall occurrence (wet/dry), and then the rainfall depth separately. Examples using Markov chains to model rainfall occurrence with probability distributions modelling rainfall depth include Richardson (1981) and Stern and Coe (1984) amongst others, and extended to the multi-site case by Wilks (1998) and Bárdossy and Pegram (2009) via Copulas. Alternatively, rainfall occurrence may also be described by an alternating renewal process, that is, sequences of serially independent wet and dry periods (Buishand, 1978), with a random variable describing the event rainfall depth. Non-parametric versions of some of the above mentioned models which sample from



empirical distributions of the modelled variables also exist (Lall et al., 1996). All of the above daily rainfall models are conceptually simple plus given the availability of observed daily rainfall data, easy to apply. These models however do not necessarily translate well to sub-daily timesteps.

Sub-daily rainfall models are often preferred, especially for flood simulation of smaller catchments. In the urban context, sub-hourly rainfall models may be required to accurately model flash floods, which are ever increasing due to land use changes and the effects of climate change. A common type of sub-daily rainfall model are point process models, which describe the arrival of storm cells in time via a Poisson distribution. Neyman-Scott type models (Rodríguez-Iturbe et al., 1987; Cowpertwait, 1991), and Bartlett-Lewis type models (Onof and Wheater, 1994; Kaczmarska et al., 2014; Onof and Wang, 2020) both model

storm cells as a collection of rainfall cells, with varying depths and durations that may overlap. The two types differ in how they describe the timing of storm cells. In Neyman-Scott models, cells are described relative to the beginning of a storm using a Poisson distribution, whereas in the Bartlett-Lewis type, the duration between cell origins is modelled via a random variable. The Newman-Scott type has been extended into space (Cowpertwait et al., 2002; Leonard et al., 2008) by modelling storm spatial extent and cell centre in space.

Alternating renewal type models, already introduced above at the daily timestep, have been also successfully applied at sub-daily timesteps (Tsakiris, 1988; Haberlandt, 1998; Bernardara et al., 2007), including the 5 minute timestep for urban applications (Callau Poduje and Haberlandt, 2017). Few attempts have been made to extend these point rainfall models in space, with the exception of Haberlandt et al. (2008). In the study, spatial consistency was applied in a two step approach. First, time series at single sites were synthesised with no consideration of neighbouring sites. Then, rainfall events were resampled

on a site-wise basis via a simulated annealing optimisation procedure conditioned on observed bi-variate spatial dependence measures. A major shortcoming however was that the method was only feasible for smaller stations networks ($\leq 6$) .

    Extending to non-rainfall climate variables, numerous parametric and non-parametric approaches exist. Richardson (1981) extended a single-site Markov based precipitation model to temperature and solar radiation using a multivariate stochastic process conditioned on the wet/dry state. Wilks (1999) improved and extended this type of model into space by using spa-

tially correlated random numbers for synthesis. More recently, Papalexiou (2018) introduced a general purpose framework to stochastically model arbitrary combinations of hydroclimatic processes at a variety of time scales.

    k-NN resampling is a flexible non-parametric approach which can easily be extended to the multi-site and multi-variate case. Cross correlations between variables are inherently maintained due to simultaneous resampling, and being non-parametric, the approach is suitable for a diverse range of climate variables. Lall and Sharma (1996) used k-NN resampling for generating

runoff time series. Daily multi-site rainfall and temperature k-NN models as by Buishand and Brandsma (2001) can further be conditioned on regional climate scenarios (Yates et al., 2003) or atmospheric circulation patterns (Beersma and Buishand, 2003). One drawback however of k-NN resampling is the inability to simulate values beyond the range of observations. Sharif and Burn (2007) overcame this limitation by introducing a random component to the output. Less common are sub-daily resampling approaches, with most in the form of method of fragments disaggregation models which resample diurnal rainfall

profiles conditioned on daily rainfall (Mehrotra et al., 2012). The intermittency of rainfall especially at the sub-daily timestep,





create challenges for resampling and Markov approaches. Hybrid approaches exist which couple stochastic rainfall models to non-parametric weather generators (Apipattanavis et al., 2007).

The present paper adopts this hybrid approach, by coupling a multi-site hourly rainfall model based on an alternating renewal approach (Haberlandt et al., 2008) to a k-NN resampling of non-rainfall variables, coupled via the daily rainfall state (wet, dry, very wet). Innovations to the multi-site rainfall model are introduced and tested at multiple scales, by applying the model to 400 meso-scale catchments of varying size across Germany. The resampling of non-rainfall variables is performed using a daily gridded dataset as observations. As a last step, the daily non-rainfall climate variables are disaggregated to hourly timesteps to match the rainfall model. Model validation is achieved by assessing the model's ability to reproduce extreme rainfall at both the site and catchment scale, its ability to reproduce observed spatial rainfall characteristics, the reproduction of observed correlations between rainfall and the non-rainfall variables, and more generally the reproduction of observed point and catchment scale statistics of both rainfall and non-rainfall variables.

## 2 Methodology

The model chain is divided into four distinct components using two primary observation sources (Fig. 1 for a model chain schematic). The foundation of the weather generator is a stochastic single-site hourly rainfall model based on an alternating renewal process (Sect. 2.1), parametrised using observed hourly station rainfall data. This single-site model is then extended into space by site-wise resampling of rainfall events via a simulated annealing optimisation approach to enforce observed spatial rainfall characteristics (Sect. 2.2). Non-rainfall climate variables are modelled using a non-parametric k-NN resampling approach (Sect. 2.3), using a gridded daily climate dataset as observations. The k-NN model is coupled to the space-time rainfall model via the catchment averaged daily rainfall state (dry, wet, very wet). Finally, to achieve the target output temporal resolution, the non-rainfall climate variables are disaggregated from daily to hourly (Sect. 2.4) using the open-source disaggregation tool MELODIST (Förster et al., 2016).

### 2.1 Single site stochastic rainfall model

The first step is the generation of synthetic hourly rainfall time series at the site level. For this, an alternating renewal model is used. Rainfall is described as an alternating sequence of independent wet and dry spells. The model shown here is a revision of the model introduced by Haberlandt (1998) and most recently further developed by Callau Poduje and Haberlandt (2017). The model consists of an internal and external structure (Fig. 2). The external structure describes the occurrence of rainfall events. The internal structure describes the temporal distribution of rainfall within a rainfall event (the event hyetograph).

The external structure is the basis of the alternating renewal model and describes the occurrence of rainfall events via the variables, wet spell duration (WSD), wet spell amount (WSA), and dry spell duration (DSD). Probability distributions are then fitted to these three event variables using the method of L-Moments (Hosking, 1990). Rainfall events are defined here as having a volume above a given threshold, the $WSA_{min}$, and a minimum separation distance to the next event of $DSD_{min}$. In this study, $WSA_{min} = 1$ mm and $DSD_{min} = 4$ hours.





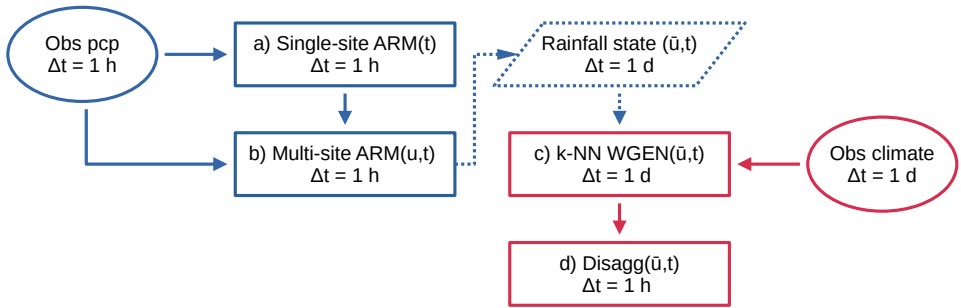

**Figure 1.** Workflow of the complete model chain. Two primary data sources are utilised, hourly station rainfall observations (left, blue) and a daily gridded climate product (right, red). The foundation and first step of the model chain is the single site stochastic rainfall model based on an alternating renewal process (a). This model is extended into space $u$ via resampling to enforce observed spatial consistence (b). Non-rainfall climate variables are resampled from catchment averaged observations using a k-NN approach (c), conditioned on the daily catchment averaged rainfall state resulting from (b). Finally, the simulated non-rainfall climate variables are disaggregated from daily to hourly (d) to meet the desired output temporal resolution.

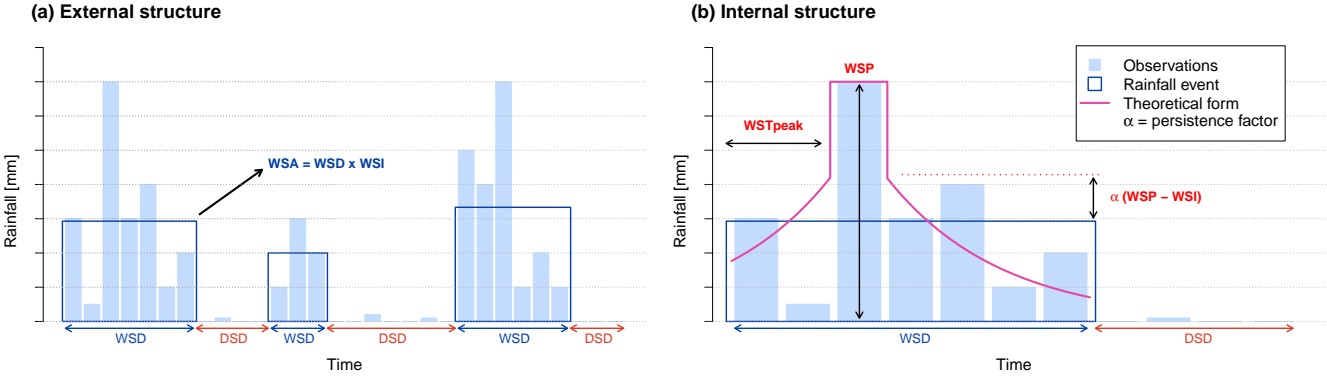

**Figure 2.** Schematic of the external (a) and internal (b) forms of the alternating renewal single site rainfall model. Observations are shown as vertical blue bars. Rainfall events are shown by dark blue rectangles. Observations falling outside of rainfall events are below the $WSA_{min}$ threshold. Events are additionally separated according to the $DSD_{min}$ threshold. For synthesis, each event is assigned an event hyetograph based on an exponential function (pink curve).





There exists however a strong dependence between the event variables WSA and WSD. Copulas are one method routinely used to model such dependencies. A copula describes a multi-dimensional space where the marginal distributions of each variable is transformed to the uniform (i.e. each dimension has the interval [0,1]). Their use in hydrology has been increasing over the years (see Chen and Guo (2019) for an overview of applications). Copulas can be of any dimension $\geq 2$, but as is the case in this study and as most frequently found in the literature and practice, only copulas of the bi-variate case will be discussed from this point onwards.

In general, a copula $C$ is a bi-variate distribution function of the uniformly transformed marginals $u$ and $v$:

$$C(u,v) = \mathrm{P}(U \leq u, V \leq v) \tag{1}$$
$$= C(F_U(u), F_V(v))$$

with $C : [0,1]^2 \rightarrow [0,1], u \in [0,1], v \in [0,1]$

The primary benefit of this transformation is that the marginal distributions of $U$ and $V$ play no role in describing the dependence between the variables.

Of special importance in this study is the modelling of extreme events with high rainfall intensity. These events by definition tend to have short duration but large rainfall depth. A copula capable of modelling these extreme events thus should be asymmetric (these events appear in one corner of the copula only). In the previous version of the model (Callau Poduje and Haberlandt, 2017), the concept of a regional empirical copula was introduced for this purpose. Within the study area, rainfall events were first normalised on a station wise basis then appended together to form a study area wide empirical copula. One drawback is that only observed events can be sampled from the copula. Events more extreme then previously observed could never be synthesised. This is particularly problematic as observation lengths are often limited due to relatively few recording stations being available at the sub-daily timestep, especially regarding longer recording periods.

For this study, a Khoudraji/Gumbel copula has been chosen to model the WSA-WSD dependence. A Khoudraji copula is a device in which two copulas $C_1$ and $C_2$ are combined via the equation:

$$C = C_1(u^{1-a_1}, v^{1-a_2}) \, C_2(u^{a_1}, v^{a_2}) \tag{2}$$

with $a_1$ and $a_2$ being shape parameters in the range [0,1]. A common use of the Khoudraji copula is to better model asymmetries (i.e. $C(U,V) \neq C(V,U)$). In its use here, an independence copula is selected as $C_1$ and a single parameter Gumbel (a.k.a. Gumbel-Hougard) copula, which shows greater dependence in the positive tail, as $C_2$. The second shape parameter $a_2$ is fixed to 1, meaning that the combined copula is described by two parameters. An independence copula is one where the uniform marginals $U$ and $V$ show no dependence (correlation = 0) and is defined by:

$$C^I(u,v) = uv = C_1 \tag{3}$$

The bi-variate Gumbel copula is defined by:

$$C_\theta^G(u,v) = \exp\left(-\left[(-\ln u)^\theta + (-\ln v)^\theta\right]^{1/\theta}\right) = C_2 \tag{4}$$





and parameterised by $\theta \in [1, \infty)$.

An additional modification from Callau Poduje and Haberlandt (2017) is the use of the Weibull distribution for both the DSD and WSA marginals in place of the Kappa distribution. As Weibull is a three parameter distribution versus Kappa's four, better model parsimony has been achieved without any significant loss of performance. This is also of benefit in any regionalisation setting. The cumulative distribution function for the Weibull distribution is defined by:

$$F(x) = 1 - \exp\left[-\left(\frac{x+\zeta}{\beta}\right)^{\delta}\right] \tag{5}$$

with $x > 0$, $\zeta \geq \min(x)$ a location parameter, $\beta > 0$ a scale parameter and $\delta > 0$ a shape parameter.

The WSD is modelled using the 3 parameter log normal distribution defined by:

$$F(x) = \phi\left[\frac{\log(x-\zeta) - \mu_{\log}}{\sigma_{\log}}\right] \tag{6}$$

with $\phi$ being the cumulative distribution function of the standard normal distribution, $\zeta$ the lower bound (real space) of $x$, and

$\mu_{\log}$ and $\sigma_{\log}$ being the mean (location parameter) and standard deviation (scale parameter) respectively of $x$ in the natural logarithmic space.

The internal model structure describes the temporal distribution of rainfall within a rainfall event, in other words the event hyetograph (Fig. 2, right hand side). The following exponential function is used to describe the rainfall intensity over time during an event:

$$P(t) = \begin{cases} \text{WSP} \cdot \mathrm{e}^{[c\lambda(t-\text{WSPT})]\alpha} & \text{if } t \neq \text{WSPT} \\ \text{WSP} & \text{if } t = \text{WSPT} \end{cases} \tag{7}$$

$$\text{with } c = \begin{cases} +1 & \text{if } t < \text{WSPT} \\ -1 & \text{if } t > \text{WSPT} \end{cases}$$

with $P$ being the rainfall intensity for timestep $t$, WSP the wet spell peak intensity, and WSPT the timestep of the wet spell peak. $\lambda$ is solved numerically for each event separately, and the WSPT is sampled from a uniform distribution. $\alpha$ is used to

adjust the shape of the exponential curve relative to the WSP (Fig. 2) and as per Callau Poduje and Haberlandt (2017) is set to 1/3.

Differently to what is presented in Callau Poduje and Haberlandt (2017), the WSP is now modelled by fitting a Weibull distribution (eq. 5) to the ratio WSP:WSA. Events with a length equal to the timestep (one hour) are first excluded. This restricts the range of possible values to between (0,1). A Khoudraji/Gaussian copula then models the dependence of the ratio WSP:WSA

to WSD. Here again, use of an asymmetric Copula via Khoudraji's device (again with a fixed second shape parameter = 1 and an independence Copula as $C_1$ as per eq. 2) showed best results. The bi-variate normal copula is defined by eq. 8. The previous model relied on a symmetrical Gaussian copula to describe the dependence between WSP and the event wet spell intensity (WSI = WSA/WSD). This approach was problematic during synthesis, as wet spell peaks could be sampled producing a wet





spell peak greater than the event WSA. The new method avoids this issue. The normal copula is defined as:

$$C_{\phi^R}^G(u,v) = \phi_R\left(\phi^{-1}(u), \phi^{-1}(v)\right) = C_2 \tag{8}$$

with $\phi_R$ being the joint bivariate Gaussian distribution function with correlation matrix $R$ for $u$ and $v$, and $\phi^{-1}$ being the inverse normal cumulative distribution function.

The rainfall event definition (requiring WSA $\geq$ WSA$_{min}$, DSD $\geq$ DSD$_{min}$) leads to a systematic underestimation of total rainfall. As a final step, small events (rainfall events below the WSA$_{min}$ threshold) are added to the time series. The method simply resamples (with replacement) small events from observations until the observed proportion of small to large rainfall events is met. Event depth, duration and distance to the next adjacent large event are sampled simultaneously and placed randomly in dry spells of adequate length.

## 2.2 Space-time rainfall synthesis via resampling

Following generation of single-site rainfall time series, rainfall events are then resampled to reproduce observed spatial rainfall dependence across a catchment. The resampling procedure is an extension of the simulated annealing optimisation approach described by Haberlandt et al. (2008). By reshuffling rainfall events (as opposed to hourly timesteps), the independence of rainfall events at single sites, as is a pre-condition of the alternating renewal model, is maintained.

Simulated annealing is a discrete optimisation procedure which is well suited to finding global minima (Bertsimas and Tsitsiklis, 1993). The method as presented here minimises an objective function describing spatial rainfall dependence by swapping rainfall events at sites at random. All reductions in the objective function are accepted, however swaps which result in an increase can also be accepted with a probability relative to the current annealing temperature. The annealing temperature decreases as the algorithm proceeds, making it less and less likely that bad swaps will be accepted. By sometimes accepting a worse objective function result, the algorithm avoids becoming stuck in local minima and aids in finding the global minimum.

As we are attempting to recreate observed spatial rainfall dependence, the objective function incorporates three bi-variate rainfall dependence criteria. The first describes the probability of simultaneous rainfall occurrence at two stations

$$P_{k,l}(z_k > 0 | z_l > 0) = \frac{n_{11}}{n} \tag{9}$$

where $n_{11}$ is the number of timesteps with simultaneous rainfall occurrence at $k$ and $l$ and $n$ is the total number of (non-missing) timesteps.

The second criterion describes the Pearson correlation of simultaneous rainfall at both $k$ and $l$:

$$\rho_{k,l}(z_k > 0 | z_l > 0) = \frac{cov(z_k, z_l)}{\sqrt{var(z_k) \times var(z_l)}} \tag{10}$$

where $z_k$ and $z_l$ are timesteps with rainfall at $k$ and $l$.

Lastly, the third criterion is a continuity measure proposed by Wilks (1998) and is the ratio of the expected rainfall at station $k$ for timesteps with and without simultaneous rainfall at station $l$.

$$C_{k,l} = \frac{E(z_k | z_k > 0, z_l = 0)}{E(z_k | z_k > 0, z_l > 0)} \tag{11}$$





A continuity close to one describes independent stations, whereas values approaching zero describe increased dependence.

In the paper by Haberlandt et al. (2008), these three spatial rainfall criteria were combined into one objective function as follows:

$$O_{k,l} = w_P \times (P_{kl} - P_{kl}^*)^2 + w_\rho \times (\rho_{kl} - \rho_{kl}^*)^2 + w_C \times (C_{kl} - C_{kl}^*)^2 \tag{12}$$

for stations $k$ and $l$ with $w_P$, $w_\rho$, and $w_C$ being weights above zero to account for the effect of differing scales between the
three criteria, and * denoting target values.

Experimentation however showed that splitting the three part objective function into two separate objective functions could lead to a faster and more optimal convergence. Of the three criteria, the occurrence criterion is by far the hardest to converge. In addition, the Pearson correlation criterion shows high sensitivity which would hamper the convergence of the other two criteria. Therefore the optimisation procedure has been split into a two-step process, where first the occurrence criterion (eq. 10) is
converged (step 1), followed together by the correlation (eq. 11) and continuity (eq. 12) criteria (step 2). For this to be possible however, in the second step, only events of equal length are swapped, in order to maintain the rainfall occurrence which has already been optimised in step 1. At low station counts the benefit of such an approach is negligible, however with increasing station count (N>20), such a method can halve the computational time required and shows increased overall performance.

In the above description of the objective function, only two stations are shown. To perform the resampling algorithm on
station networks larger than two (which is almost always the case), the objective function is calculated for all station pairs and combined using the root mean square. In the study by Haberlandt et al. (2008), stations were shuffled in sequence. That is, the first station was left unshuffled, then the second station was shuffled against the first, then the third against both the first and second and so forth. It was shown that such an approach is only effective up to station counts of around 5, as with each additional station, the resampling becomes less and less flexible due to fewer degrees of freedom.
To overcome this, a new shuffling procedure is introduced in this study, best described as a branched non-sequential approach.

Firstly, branching breaks up the station network into smaller groups. Groups consist of one primary and multiple secondary stations. Primary stations are selected in such a way that they are evenly distributed across the station network. Secondary stations are then assigned to the closest lying primary station by distance.
Non-sequential here describes the fact that stations are no longer shuffled sequentially across the entire station network, but rather within groups simultaneously. So for a given annealing temperature, swaps are performed among stations within a group at random allowing for a more flexible and rapid convergence.

In the following explanations, let $P$ be the set of all primary stations, $G$ be the set of all groups, with $g_p$ being the group containing primary station $p$ and secondary stations $S_p$. During shuffling, let $k$ be the station currently being optimised, $U$ be
the set of stations from which $k$ can be selected, and $R_k$ be the set of reference stations for which the objective function for $k$ is calculated.

The branched non-sequential procedure is as follows:





I. If the station network exceeds 8 stations, $P$ primary stations are chosen to form $N$ groups, otherwise the network is treated as a single group without primary stations (and proceeds to step IV). Primary stations are selected in a way which minimises the variance of group size and maximises the minimum distance between any two primary stations. Testing showed that a group size of 4 is ideal (on average three secondary stations to each primary station).

II. Secondary stations are assigned to the closest lying primary station by distance to form groups $G$.

III. Rainfall events amongst all primary stations are first shuffled and optimised using an objective function only containing the occurrence criterion:

$$O_k = \sqrt{\frac{\sum_{l=1}^{M}(P_{kl} - P_{kl}^*)^2}{M}} \qquad (13)$$

for station $k$, with $l$ being neighbouring stations $l = 1, \ldots, M$ taken from reference stations $R_k$. This is referred to as the primary shuffling step. Here $U = P$ and for each station $k$, $R_k = U$, $k \notin R_k$).

IV. In the secondary shuffling step, rainfall events of secondary stations are shuffled group-wise for each group $g_p$ using the same objective function as in the previous step (eq. 13). However, now $U = S_p$ and $R_k = \{S_p, p\}$, $k \notin R_k$. Rainfall events of primary stations (if any) remain fixed and are not shuffled. Within each group, additional stations outside of the group are added to the set of reference stations $R_k$ from which the objective function is calculated. This helps transfer information between the groups so that a consistent result is obtained across the entire study area. For this study, up to four additional primary stations and a total of 12 secondary stations (including the stations $S_p$) are included within $R_k$, selected by shortest distance.

V. In the tertiary shuffling step, shuffling occurs on a group-wise basis using an objective function containing the remaining two bi-variate spatial criteria:

$$O_k = w_\rho \sqrt{\frac{\sum_{l=1}^{M}(\rho_{kl} - \rho_{kl}^*)^2}{M}} + w_C \sqrt{\frac{\sum_{l=1}^{M}(C_{kl} - C_{kl}^*)^2}{M}} \qquad (14)$$

for station $k$, with $U = g_p$ and $R_k = U$, $k \notin R_k$. Important in this step is that only rainfall events of equal length are swapped, in order to avoid invalidating the occurrence objective criterion converged in steps III and IV. Unlike the secondary shuffling step, in this step primary stations are included in $U$ and in effect act no differently to secondary stations. As in step IV, additional stations outside of $g_p$ are selected for inclusion in $R_k$ (up to 24 in total, selected by shortest distance), again with the aim to transfer information between groups.

Within each of the three shuffling steps described above, the simulated annealing algorithm is as follows:

1. An initial annealing temperature $T_a$ is chosen. Experience shows that an annealing temperature causing an initial swap count of $\approx 80\%$ is optimal (Bárdossy et al., 2002).

2. If groups are defined, a group $g_p$ is sampled without replacement from all groups $G$. Otherwise the entire station network functions as a single group.





3. A station $k$ is chosen at random from all eligible stations within the set $U$ for the current group.

4. The initial objective function $O_{k,\text{prev}}$ is calculated for $k$ (eq. 14 if the tertiary shuffling step, otherwise eq. 13).

5. Two rainfall events from station $k$ within an allowed temporal distance are chosen at random to be swapped. The allowed distance (here, 6 events) allows a quicker convergence as the sensitivity of the objective function from swaps is greatly reduced.

6. An updated objective function $O_{k,\text{new}}$ is calculated to reflect the swap.

7. If $O_{k,\text{new}} < O_{k,\text{prev}}$, then the swap is accepted.

8. If $O_{k,\text{new}} \geq O_{k,\text{prev}}$, the swap is accepted with the probability $\pi$:

$$\pi = \exp\left( \frac{O_{k,\text{prev}} - O_{k,\text{new}}}{T_a} \right) \tag{15}$$

where $T_a$ is the current annealing temperature.

9. Steps 3-8 are repeated $N$ times.

10. If all groups have not yet been shuffled, the algorithm repeats from step 2.

11. The annealing temperature $T_a$ is reduced:

$$T_a = T_{a-1} \times dT \tag{16}$$

with $dT$ being the temperature reduction factor in the range $0 < dT < 1$. Generally a slow reduction in temperature is best (i.e. $dT \approx 0.98$)

12. The algorithm begins again from step 2.

13. The algorithm is stopped when the mean improvement of the objective function across all stations between temperature reductions is below a certain threshold, or if the mean objective function across all stations is below a certain target value.

### 2.3   k-NN non-parametric weather generator

Non-rainfall climate variables are modelled using a non-parametric k-NN approach. Non-parametric approaches have the benefit of not needing to assume any underlying distributions of the modelled variables. Traditional k-NN weather generators
resample target variables simultaneously for day $t$ with replacement from observations, conditioned on the previous day $t-1$. As the name implies, k-NN selects a possible $k$ candidate observations for resampling, selected by a distance metric between the feature vectors for day $t-1$ of the simulation and the candidate observations. The day following the selected observation is then inserted directly as day $t$ of the simulation. As target variables are resampled simultaneously, cross correlations between





target variables are inherently maintained. The conditioning on the previous day of the simulation aids in preserving the auto-
correlation of the target variables.

In this study, the k-NN resampling is further conditioned on the catchment averaged rainfall state $S$, which is the mechanism used to couple the space-time rainfall model to the k-NN model. Conditioning on the rainfall state aims to preserve correlations between the target variables and the already simulated catchment rainfall and is based on the method by Apipattanavis et al. (2007).

As the observed climate dataset used in this study is a gridded daily dataset, resampling occurs at the daily timestep. The catchment averaged rainfall state $S$ describes the daily areal rainfall of a catchment as either dry, wet, or very wet. The corresponding rainfall depths which describe these states are taken as the 50th and 95th percentiles of daily rainfall ($P_{\text{dry}} < 0.5$; $0.5 \leq P_{\text{wet}} < 0.95$; $P_{\text{v.wet}} \geq 0.95$ ). Rainfall acts here purely as a conditioning variable for the k-NN resampling of non-rainfall climate variables, and is not itself resampled.

For each day of the simulation, candidate days from observations are chosen using a moving window $\pm w$ around the current simulation day $t$. For example, if $t$ is June 15th and $w = 7$ days, only observed days between June 8th and June 22nd (from any year) may be chosen. This allows for the reproduction of seasonal climate characteristics.

Potential neighbours are then further reduced by conditioning by rainfall state $S$ for both days $t$ and $t - 1$ of the simulation. For example, if simulated day $t$ is wet and simulated day $t - 1$ very wet, only observed days which are very wet followed
by a wet day (and within the observation window $\pm w$) may be chosen. If no days from observations match this criteria, this conditioning is relaxed to apply to day $t$ only.

Feature vectors $D$ are created for each day of observations, with $D$ consisting of the normalised catchment averaged variables $x'$. Each climate variable $x$ is first normalised by subtracting the mean and dividing by the standard deviation:

$$x' = \frac{x - \overline{x}}{\sqrt{\text{var}(x)}} \tag{17}$$

The k-NN procedure proceeds as follows:

a. For the first day of the simulation $t = 1$, an observed day within the selection window $\pm w$ is selected at random conditioned only on the rainfall state $S_t$.

b. The simulation day $t$ is advanced by 1.

c. Observed days within the selection window $\pm w$ are selected to form candidate days $U$.

d. $U$ is reduced by conditioning on the rainfall state $S_t$ and $S_{t-1}$.

e. A distance metric, the weighted euclidean distance $\delta(D_{t-1}, D_u)$, is calculated for each day $u$ in $U$ and simulation day $t - 1$:

$$\delta(D_{t-1}, D_u) = \left[ \sum_{j=1}^{N} w_j (x'_{t-1,j} - x'_{u,j})^2 \right]^{1/2} \tag{18}$$





with $w_j$ being the weight for climate variable $x'_j$, $N$ the total number of climate variables, $x'_{t-1,j}$ and $x'_{u,j}$ being the normalised climate variable for day $t-1$ and candidate day $u$. For this study, variable weights were assigned manually by trial and error. A higher weight for one variable over another will generally improve the performance of that variable regarding its correlation to rainfall.

f. Candidate days are then ordered from nearest to farthest and given ranks $j$.

g. $U$ is then further reduced to $k$ neighbours based on lowest rank (closest distance). The selection of $k$ is user definable, but is often taken as $k = \sqrt{N}$ where $N$ is the sample size, as proposed by Lall and Sharma (1996). As the window size $\pm w$ restricts possible neighbours, $k = \sqrt{Y \times (2w+1)}$, with $Y$ equal to the length of observations in years.

h. A single day is then selected from $U$ using a discrete probability distribution. Lall and Sharma (1996) recommended a kernel which gives increased weight to nearer neighbours:

$$p_j = \frac{1/j}{\sum\limits_{i=1}^{k} 1/i}, \text{ for } j = 1, \ldots, k \tag{19}$$

i. Day $t$ of the simulation is then taken as day $u+1$.

j. The algorithm begins again from step b until all days have been simulated.

The choice of which combination of climate variables to resample depends largely on the intended end-use of the simulation and the availability of observations. As the feature vector $D$ contains normalised climate values, variables of any magnitude and distribution may be used. Heavily skewed variables may undergo an optional log transformation. For this study, relative humidity, temperature (daily mean, minimum and maximum) and global radiation were chosen, as the intended end-use is derived flood frequency analysis using the hydrological model HBV (Lindström et al., 1997). HBV requires rainfall, temperature and potential evaporation as input, which can all be directly used or derived from these chosen variables. Furthermore, due to ease of use and availability, a gridded observational climate dataset was used for this study. However the k-NN resampling approach presented is not limited to gridded datasets and may be applied to networks of point observations, with the feature vector $D$ representing mean values across the station network.

## 2.4 Disaggregation from daily to hourly

A final step is required to disaggregate the resampled daily non-rainfall climate variables from daily to hourly. This is achieved using the open-source disaggregation tool MELODIST (Förster et al., 2016).

Temperature is first disaggregated to hourly values $T_{d,h}$ for day $d$ and hour $h$ using the cosine function (Debele et al., 2007):

$$T_{d,h} = T_{\min,d} + (T_{\min,d} + T_{\max,d})/2 \times (1 + \cos(\pi(h+a)/12)) \tag{20}$$





with $T_{\min,d}$ and $T_{\max,d}$ being the minimum and maximum temperatures for day $d$. The parameter $a$ describes the time difference between solar noon and the time of maximum daily temperature. For this study, $a$ is simplified to two hours across the entire year.

Humidity relies on already disaggregated hourly temperature data. Relative humidity for hour $h$ for day $d$ is calculated by:

$$H_{d,h} = 100.\frac{e_s(T_{\mathrm{dew},d})}{e_s(T_{d,h})} \, [\%] \tag{21}$$

with $e_s$ being the saturation vapour pressure of a given temperature $T$ [°C], given by the Magnus formula (Alduchov and Eskridge):

$$e_s(T) = \begin{cases} 6.1078 \exp\left(\frac{17.08085T}{234.175+T}\right) & T \geq 0°\mathrm{C} \\ 6.1071 \exp\left(\frac{22.4429T}{272.44+T}\right) & T < 0°\mathrm{C} \end{cases} \tag{22}$$

The dew point temperature is simplified and taken as the daily minimum temperature ($T_{\mathrm{dew},d} = T_{\min,d}$) and is constant throughout the day (no diurnal profile). Finally global shortwave radiation is disaggregated from daily values using a simplified formula which assumes a flat surface (Liston and Elder, 2006):

$$R_0 = 1370\,\mathrm{Wm}^{-2}.\cos Z_{d,h,\phi}.(\psi_{\mathrm{dir}} + \psi_{\mathrm{dif}})\left[\mathrm{Wm}^{-2}\right] \tag{23}$$

    with $\psi_{\mathrm{dir}}$ and $\psi_{\mathrm{dif}}$ being the direct and diffuse radiation scaling values, $Z_{d,h,\phi}$ being the local solar zenith angle for day $d$,

345 hour $h$ and latitude $\phi$, which for this study is taken as the mid-point of each catchment. Further details of the radiation scaling values can be found in Liston and Elder (2006).

## 3   Study area and data

400 meso-scale catchments in Germany were selected for this study. These catchments range in size from 30 km$^2$ to over 20,000 km$^2$. Figure 3 shows a boxplot of catchment by area.

As observed rainfall, 699 point sub-daily recording stations at the hourly timestep were sourced from the German Weather Service (DWD). A common time period of January 2006 - December 2020 was chosen to maximise station availability over the period across all stations. Figure 4 displays the location of both catchments and rain gauges.

    Stations were assigned to catchments, and the space-time weather generator is applied on a catchment basis. The largest catchment contains 87 stations, with 109 catchments containing at least 10 stations, and 27 catchments containing at least 30

stations.

    The HYRAS (Razafimaharo et al., 2020) gridded (5 km × 5 km) daily observational climate dataset was chosen for use for the non-rainfall climate variables. Climate variables include the mean, maximum and minimum daily temperature, relative humidity, and global radiation. Coverage of this dataset is German wide extending into neighbouring countries except Czech Republic. This results in a few catchments with boundaries extending into Czech Republic not having 100% pixel coverage.

For use in the k-NN weather generator, catchment averages were calculated for each catchment and climate variable. The time period 1976 - 2015 (40 years) was chosen to increase the total number of days available for resampling.





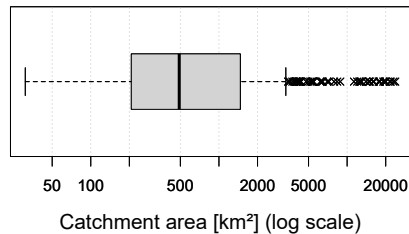

**Figure 3.** Boxplot of catchment area (N = 400).

To validate the weather generator results, 40 sub-daily reference climate stations were chosen across Germany (Fig. 4). Of these, 20 also include data for global radiation.

The climate of the study area is generally temperate, tending towards continental in the east and south east and oceanic in the north. Convective precipitation is typical in the summer months, being the primary reason why the parametric models presented here have been applied to summer and winter separately. Annual rainfall sums range from around 500 mm to over 2000 mm in southern elevated regions. Rainfall is common year round, being highest in summer and lowest in spring. Elevations are typically below 1000 m above sea level except in the southern Alpine region. Large temperature gradients are not present, with the warmest area being the Rhine valley along the border with France.

## 4 Model setup and validation

To adequately test the performance of the complete model chain, 100 realisations of 15 years simulation length were generated. 15 years was chosen to match the observation length of the sub-daily rainfall stations. The model is conditioned on summer (Apr-Sep) and winter (Oct-Mar) seasons separately.

Target values as used in the objective function of the simulated annealing resampling procedure were calculated as follows. For each catchment, the closest 100 (at least) to 150 (at most) stations from the centroid of the catchment were selected (this may also include stations located outside of the catchment boundary). For each station pair the three bi-variate spatial rainfall criteria were calculated from observations. Regression curves (not shown here) were then fitted to the observed data with station distance as the independent variable. The target values for simulations were then taken from these curves with added noise equal to the residual variance.

For the evaluation criteria, relative bias is calculated as follows:

$$Bias = 100 \times \frac{X_i^* - X_i}{X_i} \ [\%] \tag{24}$$

where $X_i$ is the observed value of the variable in question for station $i$ and $X_i^*$ the simulated value. A positive bias indicate overestimation, a negative bias underestimation.

The performance of the weather generator is evaluated as described in the sub-sections below.





**Figure 4.** Study area showing location of DWD rainfall stations (N=699), DWD climate stations (N=39) and catchments (N=400). Catchments may be overlapping.





## 4.1 Point rainfall model

The point rainfall model (Sect. 2.1) was applied in two modes for all 699 rainfall stations. The first mode is the model as described by Callau Poduje and Haberlandt (2017) and will be referred to as the 'Previous' model. The second mode incorporates the changes as introduced in this paper, and is referred to as the 'Revised' model. The two modes allow us to directly assess whether changes to the rainfall model have indeed increased its performance.

Note however that as the previous model has a target output timestep of 5 minutes, it may perform less well at an hourly timestep.

The performance is assessed via:

- Relative bias of annual precipitation sum and number of events. The median bias for each station over 100 realisations is taken.

- Relative bias of the event variables WSA, WSD, DSD, WSI and WSP. Note that WSI is indirectly modelled but acts as a good indicator of the performance of the bi-variate copula $C(\mathrm{WSA}, \mathrm{WSD})$. The median bias for each station over 100 realisations is taken.

- Extremes are assessed via the relative bias in fitted Intensity Duration Frequency (IDF) curves. IDF curves were fitted to observed and simulated ($100 \times 15$ years) annual maxima series using the robust method according to Koutsoyiannis et al. (1998) for the storm durations 1, 3, 6, 12, 24 and 48 hours. For each storm duration, the rainfall depth for a return period of 20 years was calculated. The median bias across realisations is then calculated for each station and presented in box plots for each storm duration.

- The wet/dry intermittence of daily rainfall, first by considering wet day frequencies and wet-wet/dry-wet transition probabilities (and by implication their complements, wet-dry/dry-dry transition probabilities). These are presented in observed versus simulated plots. A threshold of $\geq 0.1$ mm was used to classify wet days.

## 4.2 Space-time rainfall model

After the generation of point rainfall, the model was extended into space on a catchment-wise basis by applying the simulated annealing resampling approach described in Sect. 2.2. Due to computational constraints, the previous model is not considered here. The performance in space is assessed via:

- Spatial dependence of hourly rainfall via the three bi-variate criteria (occurrence, correlation and continuity) presented as a 2D density plot. To produce the empirical densities, for each station pair the median result across all realisations was taken.

- Like for the point rainfall model (see above), the bias in fitted areal IDF curves, again incorporating the storm durations 1, 3, 6, 12, 24 and 48 hours with a return period of 20 years, was calculated for each catchment and presented as boxplots. Catchment rainfall was calculated via the Thiessen polygon method.





## 4.3 Non-rainfall climate variables

The non-rainfall climate variables are first assessed at the daily timestep to isolate errors stemming from the k-NN resampling. The performance for catchment averaged values is assessed via:

– Summary statistics of modelled climate variables comparing mean monthly observed vs. simulated values.

– For each of the modelled climate variables, the daily auto-correlations up to lag 5 is shown plotted on observed vs. simulated plots.

– Daily correlation between rainfall vs. non-rainfall climate variables plotted on observed vs. simulated plots, shown by month.

Also of interest is how well the k-NN resampling approach performs considering point observations. For this, 39 reference
weather stations were taken from the German Weather Service observation network and compared to first, a) the grid cell values taken directly from the HYRAS observational dataset in order to first assess bias resulting from the gridded dataset, and secondly, b) the k-NN resampled values by taking the median result over $37 \times 40$ year simulations. Monthly mean values are shown on observed vs. simulated plots.

Finally, the bias due to disaggregation from daily to hourly is assessed by comparing hourly means of each climate variable
averaged across each reference station. As the intended use of the weather generator is for applications such as derived flood frequency analyses, the performance of the hourly non-rainfall variables is of lower priority.

## 5 Results and discussion

### 5.1 Point rainfall model

Figure 5 shows violin plots of the median bias of event variables, the number of annual events, and the total annual precip-
itation sum. For the event variables WSD, DSD and WSI, the previous and revised models perform almost identically. This demonstrates that the change from the four parameter kappa distribution to the three parameter log normal distribution for the variables WSA and DSD has no negative consequence on model performance. Of the directly modelled event variables, DSD shows the worst performance, and more so for summer, indicating that an alternative distribution function may be more appropriate.
The revised model shows decreased performance regarding wet spell intensities. Wet spell intensities are indirectly modelled via the WSA:WSD copula. The previous model implements a so called regional empirical copula, which resamples from observations. As the median bias shown in the violin plot directly compares simulated verses observed values, it may be that this statistics favours the previous model due to this resampling. On the other hand, the revised model shows a substantially better modelling of the wet spell peak, which validates the new wet spell peak modelling approach. Except for WSP and WSI,
the bias lies within $\pm10\%$ range.





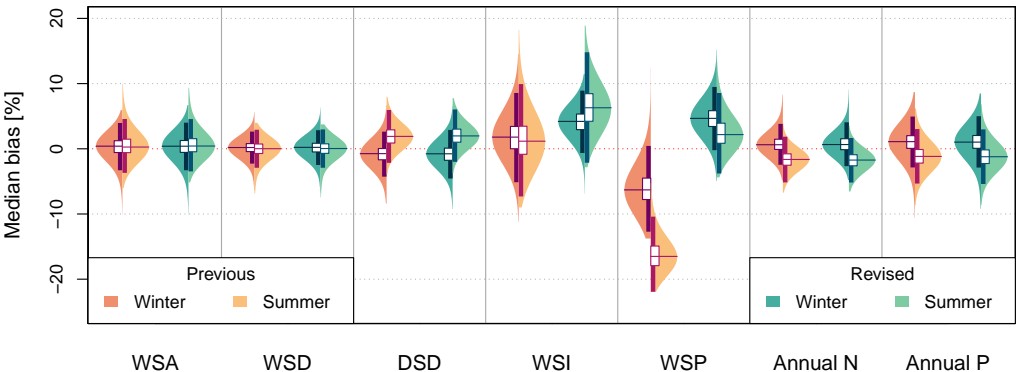

**Figure 5.** Violin plots of the median bias from $100 \times 15$ year simulations for 699 rainfall stations. Left side of violin plot shows results for winter, right side for summer. Annual N refers to the mean number of annual rainfall events and Annual P the mean annual rainfall sum (before the addition of small events).

Finally, in terms of annual number of events and rainfall sum, both models perform similarly well, with summer showing a greater underestimation than winter shows overestimation. This is likely due to a mean overestimation of DSD in summer as was shown above. Here bias also lies within $\pm 10\%$ range.

Figure 6 shows the bias for extreme rainfall, split by season. In general it can be seen that the revised model shows a far
better performance. Most storm durations show a median result close to zero, however winter at the 1 hour storm duration shows an overestimation and large range ($\pm 30\%$). With increasing storm duration, an increasing underestimation is seen. The previous model significantly underestimates extreme rainfall, which is likely caused by the significant underestimation of wet spell peaks.

Figure 7 shows wet/dry day statistics for all stations for both summer and winter seasons and for both the previous and
revised models. Large differences between the previous and revised model are not seen, so the points discussed here apply to both model versions. The relative frequency of wet days is generally well maintained with no obvious bias for both summer and winter. Dry-wet day transition probabilities show in winter a greater underestimation (and conversely an overestimation of dry-dry day), however with a mean underestimation of only $\sim 5\%$. Wet-wet day transition probabilities are generally overestimated. Overall both models show a good reproduction of all wet/dry day statistics.

**5.2 Space-time rainfall model**

Figure 8 shows the performance of the three bi-variate spatial criteria in the form of 2D density plots over all catchments and station pairs. Station separation distances of up to 150 km are shown. Results for the occurrence criterion show an overall good reproduction of observations with no significant loss, however with a narrower range of values, especially for summer. As the occurrence criterion is optimised first before the other two criteria, we should expect good results. For Pearson's correlation,





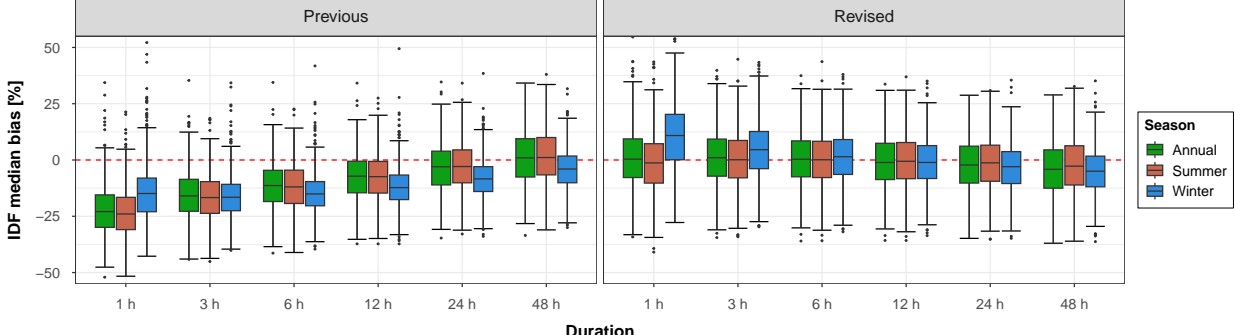

**Figure 6.** Boxplots of median IDF bias over 100 realisations for all stations across the study area (N = 699) for various storm durations, split by season, for both the previous and revised models. A return period of 20 years has been used.

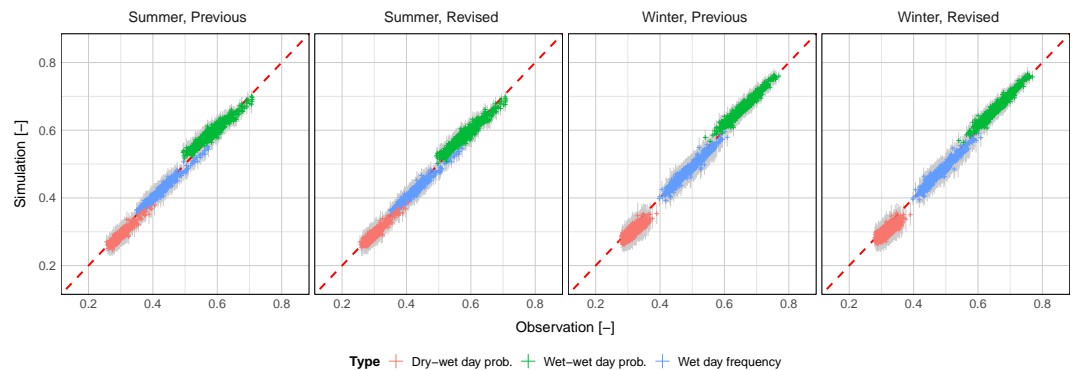

**Figure 7.** Daily wet day statistics for all stations across the study area (N = 699), for both the previous and revised models, split across seasons. Grey lines show the range of results across 100 realisations. The dry-dry and wet-dry transition probabilities, and the dry day frequency, can be inferred from the plots using the complement.

the general form of the density plot is maintained but again with a narrower range of values. Above 75 km there is a greater loss in performance unlike for the occurrence criterion. This is most likely due to the fact that such distant stations are not included in the objective function of the re-sampling optimisation approach, in order to reduce computational complexity. Lastly the continuity criterion can be described as the worst performing of the three criteria, especially regarding summer. A general loss in performance can be seen across all station distances, however the general form of the density plot is maintained.

The reproduction of extreme catchment rainfall has been assessed via the bias in areal rainfall depth for a return period of 20 years for varying storm durations as shown in Fig. 9. For the one hour storm duration, winter performs significantly worse than for summer, especially regarding the median and range. This matches the result seen at the station level. The median annual result is close to zero however with a wide overall range ($\pm 30\%$). From duration 3 hours and above, winter performs similarly than for summer. A general underestimation ($\sim 10\%$) of extreme rainfall can be observed.

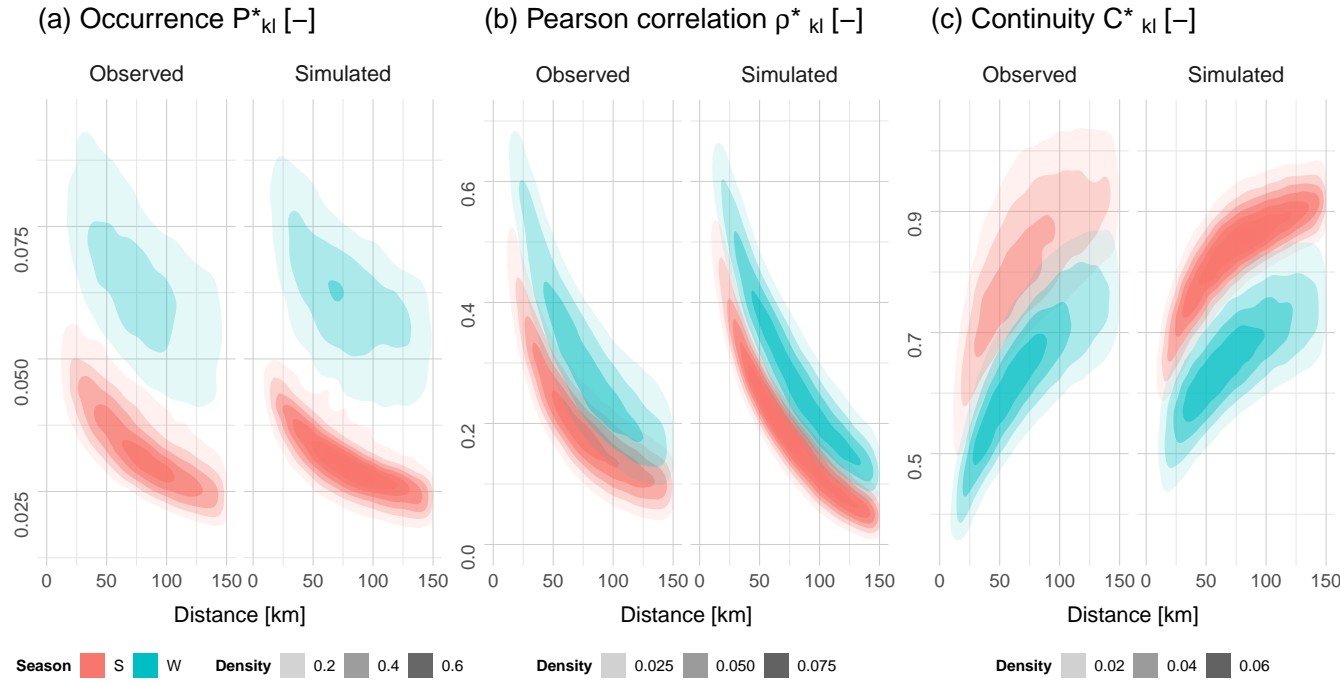

**Figure 8.** 2D density plots of the occurrence (a), Pearson's correlation (b) and continuity (c) bi-variate spatial dependence criteria, grouped by season, for all station pairs across all catchments (N = 19,773), by station distance. For simulated results, the median value from 100 realisations was taken.

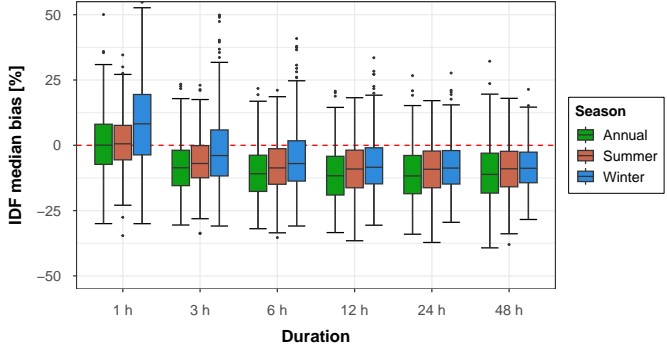

**Figure 9.** Boxplots of median areal IDF bias over 100 realisations for all catchments across the study area (N = 400) for various storm durations, split by season. A return period of 20 years has been used.





## 5.3 Non-rainfall climate variables

The non-rainfall climate variables were first assessed at the daily timestep (before disaggregation to hourly), to isolate errors arising from the k-NN resampling procedure.

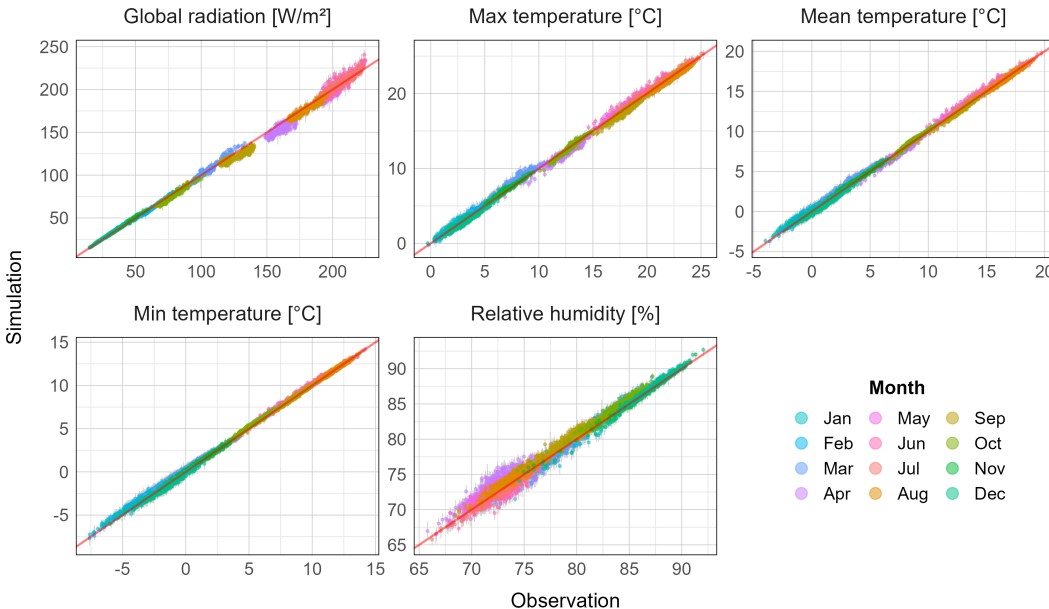

**Figure 10.** Scatter plots of mean monthly values for all catchments (N=400). The range from all realisations is shown via grey bars and the median is shown coloured by month.

Mean monthly values of all resampled climate variables are shown in the form of observed vs. simulated scatter plots (Fig. 10). All three temperature variables show a good reproduction over the year, with no systematic under- or overestimation. Global radiation performs worse in both spring and summer, with the highest values showing greatest spread. The relative absolute bias is rarely greater than 5% however. Relative humidity shows the greatest spread of values, particularly for the months April-July. For all variables, large variations across realisations was not shown.

The ability of the k-NN resampling procedure to maintain observed variable auto-correlation is shown in Fig. 11. The magnitude of auto-correlation is well preserved for all variables and all lags, however as expected a certain loss in auto-correlation is seen, particularly for lag 1. In general it can be said that the temperature variables are the better performing. The auto-correlation results are sensitive to the weights used in the distance metric (eq. 18). As these are user assigned, the performance relating to auto-correlation can be manipulated to a degree. The results here also show only small variations across realisations for all variables.

The Pearson correlation between resampled climate variables and daily rainfall is shown in Fig. 12. Also here cross-correlation is reproduced well, however a loss in correlation is observed for all variables across most months. The temper-





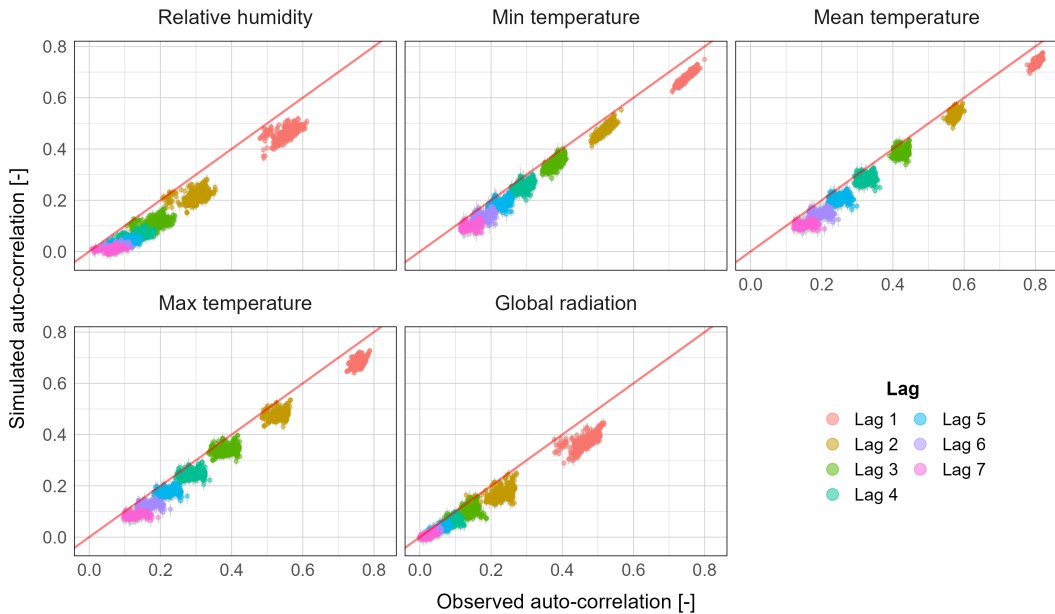

**Figure 11.** Scatter plots of variable auto-correlation for lags 1 to 7 for all catchments (N = 400). The range from all realisations is shown via grey bars and the median is shown coloured by month.

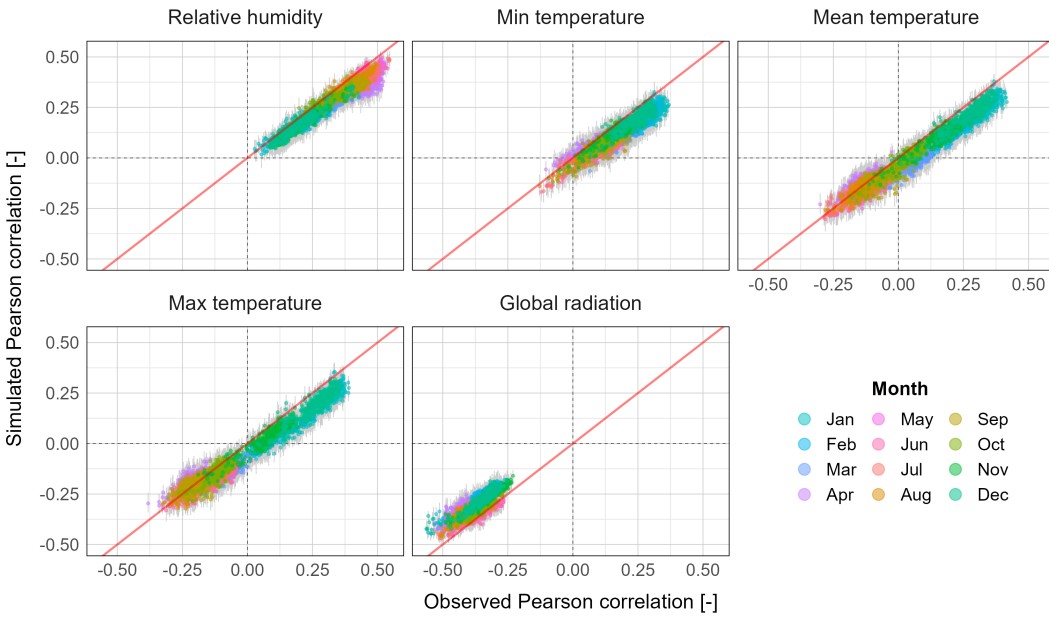

**Figure 12.** Scatter plots of correlation to daily rainfall for all catchments (N = 400). The range from all realisations is shown via grey bars and the median is shown coloured by month.





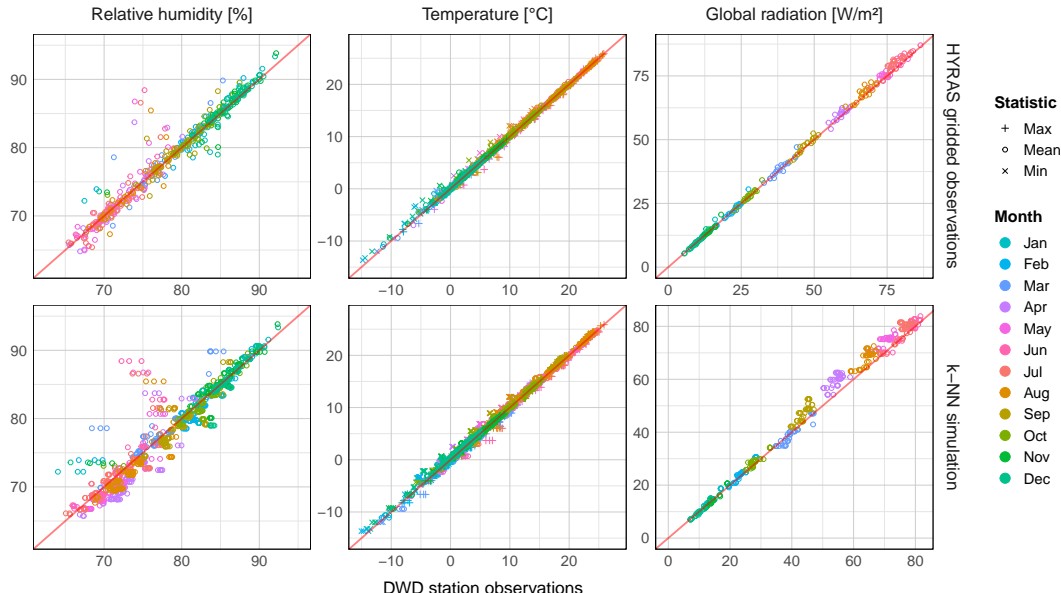

**Figure 13.** Scatter plots showing bias of mean monthly values between point weather station observations and, (top) HYRAS gridded observations for the same time period (1976-2015), and (bottom) median bias from $37 \times 40$ year simulations after k-NN resampling. As weather stations can overlap multiple catchments, more data points are included in the bottom plots. Temperatures were corrected for elevation difference between the grid cell elevation and weather station elevation using a lapse rate of $6.5^\circ\mathrm{C.km}^{-1}$

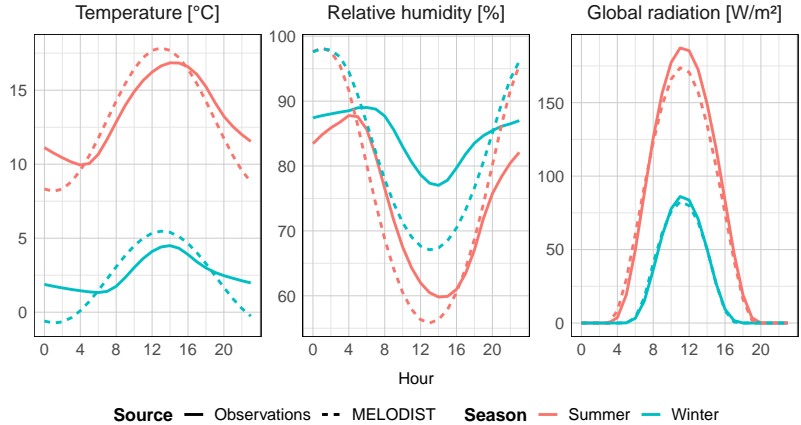

**Figure 14.** Mean hourly values of the non-rainfall climate variables, separated by season for 39 DWD reference stations. Simulations show output from the MELODIST disaggregation model, using daily observations as input.





ature variables are worst performing during the winter months, where correlation is highest. Summer months where a smaller negative correlation is observed, perform on average better however with a greater spread of results. Relative humidity also performs worse for summer months. Correlation to global radiation is seen to be less dependent on month and a general loss of correlation is observed.

To assess errors stemming from the use of the gridded climate dataset, observed daily time series from 39 reference stations from the German Weather service where compared against corresponding HYRAS gridded values both before and after resampling (Fig. 13). Looking at the results before resampling (top row of figure), performance is generally good with absolute bias generally not exceeding 2°C for temperature. Radiation shows a very good reproduction of observations, however relative humidity shows in part a very poor reproduction. The results after resampling (bottom row of figure) mimic those of before,

however with a small increase in bias, with temperature still performing very well, followed by global radiation and humidity. This shows that the use of the gridded dataset does not bring about a significant loss of performance, with the exception for relative humidity.

   Finally the disaggregation to hourly performance is assessed by comparing hourly observed vs. simulated means of each climate variable, as shown in Fig. 14. Both temperature and global radiation exhibit a timeshift of one hour. This could be

manually corrected in future revisions. Significant deviations between observed and simulated values can be seen, particularly for daily minimum values, and for relative humidity in winter. However as the model's intended end-use is derived flood frequency analyses, the recreation of diurnal profiles is of lower priority. In contrast to other results, global radiation is the best performing of the three resampled variables.

## 6   Conclusions

This study presents a major revision of the previous space-time rainfall model by Haberlandt et al. (2008). Large station networks of over 80 stations can now be modelled with limited loss in the observed spatial dependence structure. This was achieved by introducing a novel multi-site branched non-sequential event resampling approach based on a simulated annealing discrete optimisation procedure, extending the single site rainfall model into space. Further modifications to the single site rainfall model also resulted in improved model parsimony and performance regarding rainfall extremes. The replacement of the

previous empirical Copula for modelling event wet spell intensities with a theoretical copula allows for future regionalisation applications.

   Furthermore, the coupling of the space-time rainfall model to a non-parametric k-NN weather generator and subsequent disaggregation now provides the user a single tool for the generation of hourly climate time series for applications such as derived flood frequency analyses. Coupling the two sub-models via rainfall state allowed an adequate reproduction of both

auto-correlations and correlation to rainfall. The flexibility of the approach allows the modelling of a diverse range of climate variables and observation sources (point or gridded).



By testing the complete model on 400 catchments and 699 rainfall stations across Germany, the model was shown to perform across a wide range of catchment sizes and locations. Future studies may assess the performance in different climates and more diverse terrain.

Currently the space-time rainfall model is run separately for summer and winter seasons. This rudimentary partitioning is one potential area for future improvement. Conditioning the model on circulation patterns, which better categorise different rainfall and weather pattern types, may lead to increased performance, particularly regarding extremes. It may however be that a conditioning of the model on circulation patterns is too restrictive, especially as observation lengths of sub-daily rainfall are still generally too short.

*Author contributions.* Supervision and funding for this research were acquired by UH, the study conception, design and methodology were performed by both authors, while the software development, data collection, derivation and interpretation of results were handled by RP. RP prepared the original draft, which was revised by UH.

*Competing interests.* The authors declare that they have no conflict of interest.

*Acknowledgements.* The authors gratefully acknowledge the financial support of the German Research Foundation (Deutsche Forschungs-
gemeinschaft, DFG) in terms of the research group "Space-Time Dynamics of Extreme Floods (SPATE)" (grant no. FOR 2416), and the German Weather Service (Deutscher Wetterdienst, DWD) for providing the gridded observational dataset HYRAS and climate station data.





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
