# Peer review of "A semi-parametric hourly space-time weather generator"

_Hydrology and Earth System Sciences, 2023_

## Author Comment (AC1)

**Response to reviewer #2 (anonymous)**

The idea of the article is original (development of a space-time rainfall model that can accurately reproduce various rainfall statistics including extreme values) and is also very well organized.

**Response:** I thank the reviewer for their supportive feedback.

L89: Is there any mechanism in your model to consider temporal autocorrelation of WSD, WSA, DSD while generating them? (Large rainfall is quickly followed by large rainfall, and vice versa) This mechanism will enable the model to reproduce long(er)-term rainfall variability (e.g. weekly, monthly) making it more versatile (Kim et al., 2020).

**Response:** At this stage there is no mechanism to consider event variable auto-correlation. As event variables are randomly sampled from fitted probability distributions, a possible method to model autocorrelation would be to include auto-correlation in the simulated annealing optimisation procedure, which is currently utilised to enforce spatial consistence. It must however by said, that observed event variable auto-correlation is relatively low. Median lag-1 auto-correlation across all 699 stations is 0.0269 for wet spell amount, 0.0822 for wet spell duration and 0.0482 for dry spell duration,

L147: Why Weibull? Generalized Pareto Distribution may be a better pdf for rainfall peak values. You may want to try the L-moment diagram method to figure out the most optimal distribution of WSP.

**Response**: We thank the reviewer for the comment and suggestion. The below diagram shows the L-moment diagram for all stations used for the study. It indicates that the Generalized Pareto Distribution may well be a good choice for the variable WSP:WSA.

[Figure]

To test its suitability, the WSP was modelled over 100 realisations of 15 years using both Weibull (as in the study) and GPA distributions (as suggested by the reviewer). Looking at the absolute relative bias, abs(mean(sim) – mean(obs) / mean(obs)), with the median taken over all realisations (below figure), a slight improvement can be observed in winter, however performance is slightly worse for summer. Other metrics (e.g. 98% percentile, standard

deviation etc) align with this finding, and as such it is not considered worth the effort to change the distribution at this late stage.

[Figure]

Section 2.2. Space-time rainfall synthesis via resampling: I have an impression that the model is too much oriented toward reproducing only spatial-correlation. Do you have any algorithm to ensure space-time correlation at all gauges? The algorithm does not seem to have a capacity to simulate continuous movement of storms. In other words, do the consecutive snapshot of rainfall fields resemble with each other?

**Response:** The current model is only able to reproduce spatial consistency dependent on inter-station distance alone. Direction is not considered, nor are any temporally lagged values incorporated into equations 9-11, which could aid in modelling moving storm fronts. The focus of this paper (and where most of the effort occurred) is its multi-scale performance to model both small and large rainfall gauge networks, as previous attempts had failed to do so. The nature of the simulated annealing optimisation procedure however allows for future revisions to incorporate additional characteristics such as storm front movement etc if the intended use of the output time series warrants it.

Figure 11. The systematic underestimation may be related to the first comment of this review.

**Response:** The k-NN resampling method, unlike the space-time rainfall model, aims to reproduce daily auto-correlation of the non-rainfall climate variables. The mechanism to achieve this is via the distance metric (equation 18) used for the selection of the k nearest neighbours. There is expected to be a systematic underestimation in auto-correlation, as the best ranked candidate day is not necessarily chosen due to the discrete probability distribution used (equation 19) and the conditioning on rainfall state further restricting possible candidates.

I suggest authors to add another figure that shows a diagram showing the space-time autocorrelation between Figure 8 and Figure 9.

**Response:** By space-time autocorrelation I assume that the reviewer is referring to lagged correlation between stations by station separation distance? As mentioned in R2C4 above, as lagged correlations were not a focus of the study, such a figure or discussion has not yet been included. Performance in this regard is unsurprisingly weak, due to not being included in the

objective function. The preliminary plot below shows the first 4 lagged correlations. Such a figure could be included in the work if requested, however I question the value in doing so.

---

## Author Comment (AC2)

**Response to reviewer #1 (Simon Michael Papalexiou)**

This is a useful paper and deserves consideration. I will avoid a generic summary and list right away several comments that hopefully the authors might find useful.

**Response**: I thank the reviewer for their many constructive and insightful comments.

As a general comment, the methods described suit more multisite methods, especially for precipitation. At the hourly or finer scales, space-time precipitation has components or advection and anisotropy, and it is not clear if what these methods preserve in this regard (I wil get back to that).

**Response**: It is true that the model presented does not preserve any attributes regarding advection or anisotropy. The focus of the research was predominantly developing methods that allow for the synthesis of large stations networks (station networks up to 100), as previous versions of the single site alternating model could only achieve this to a limited extent (station networks less than 10 – see Haberlandt et al. 2008). Anisotropy could be considered in future revisions by adding direction as an independent variable within the objective functions used in the simulated annealing optimisation approach. Advection may also be considered by considering temporally lagged values of one or more of the three bi-variate rainfall dependence criteria. This topic will be discussed further in responses below.

1. please see also Papalexiou (2022) which is dedicated only to precipitation.
   **Response**: Thank you for the reference. The introduction will be updated accordingly.
2. Rainfall is described by the single-model as an alternating sequence of independent wet and dry spells. How valid this assumption is?
   **Response**: Observed auto-correlation of the event variables is low. For instance, the median lag-1 auto-correlation across all 699 stations is 0.0269 for wet spell amount, 0.0822 for wet spell duration and 0.0482 for dry spell duration.
3. I can understand the 1 mm limit but based on what rationale the DSDmin = 4 hr was set?
   **Response**: Choosing 4 hours for the DSDmin value was a somewhat arbitrary decision. The previous study by Callau et al (2017) used a value of 1 hour. The German guideline for calculating return periods of extreme rainfall events (DWA-A 531) requires a dry period of at least 4 hours between events to be considered independent. The text will be updated to include more background as to this choice.
4. if u,v are in [0,1] then where are $F_U(u)$ and $F_V(v)$. Check the notation please.
   **Response**: Thankyou for the comment. Notation will be revised.
5. What is the justification of this choice? Have you tested e.g., the Gumbel dependence and was not suitable? Did you observe asymmetries? Please justify.
   **Response**: I assume the reviewer here is referring to the choice of copula? The main driver in the choice of Khoudraji's device, is the presence of asymmetries in the observations (events with large WSA, but small WSD). As to the choice of the Gumbel copula for $C_2$, many copula families were tested by trial and error, and Gumbel was found to lead to the best performance regarding wet spell intensities. A better explanation regarding the choice of copula will be included in the revised paper.
6. Similarly, what led you to the Weibull choice for DSD and WSA. Great that you've drop the 4-parameter Kappa but why Weibull is a good model for the WSA. If you consider that, e.g., the hourly wet value distribution is a specific distribution then then

WSA would be its convolution. Specifically, for the Weibull there were some attempts to justify it theoretically as a rainfall distribution by Wilson & Toumi (2005); it was also used in meta statistical approaches for daily rainfall e.g., (Marani & Ignaccolo, 2015; Marra et al., 2018, 2023) and it seems it does a good job in describing the extremes but if it is suits well for the WSA it will be nice to show some evidence. Also, here you're using the 3-par version which also can end up with $\zeta$ quite larger than the min so you might have inconsistencies in low values. Can you explain please? The same point holds for the distribution choice of the WSD. Is the LN supported by the literature? By your own analysis in this dataset?

**Response**: Like for the previous response, the Weibull was chosen more through trial and error than any rigorous theoretical foundation. Many distributions were explored, then goodness of fit criteria such as the Cramer-von-Mises test and visual tests such as QQ-plots were used to arrive at the choice of Weibull for the DSD and WSA and LN for the WSD. If requested, a section can be included to discuss the choice in more detail and perhaps provide some goodness-of-fit results. Also, sometimes the lower bounds of the distributions falls below the WSAmin or DSDmin, in which case any simulated value below these thresholds are replaced with the threshold value. The text will be updated to mention this.

Equation 7. So this implies that the tail of intensity is exponential? It could for this region but in general this contradicts many global studies indicating that the tails are not exponential but heavier. Exactly for this reason, in the past I explored the Generalized Gamma to allow heavier right tails and recently some Generalized Exponential distributions having similar tails (see Papalexiou, 2022). I believe the choice of model is crucial as we're risking underestimating the potential for extremes. Please explain.

**Response**: Equation 7 describes the distribution of rainfall within the event. It should be emphasised that the internal distribution of rainfall within an event is considered of lower importance due to the intended end use of derived flood frequency analysis of meso-scale catchments where an exact reproduction of the time series over small time scales isn't necessarily required. The wet spell peak of an event is modelled through a copula that describes the dependence between the ratio WSP:WSA and the wet spell duration (eqs 2, 3, 8) and the Weibull distribution to describe the ratio WSP:WSA. Equation 7 only applies to the timesteps before and after the wet spell peak (see figure 2 for a visual representation). The text will be modified to make this clearer to the reader.

Also what is the correlation structure within the event? Clearly at hourly resolution there is strong autocorrelation within wet values (see Papalexiou, 2022).

**Response**: Auto-correlation is generally over-estimated in simulated values (see below plot of observed and simulated auto-correlation for all 699 stations up to lag 5). A plot and

discussion of the correlation structure can be included in the final manuscript if required.

[Figure]

1. lower phi is typically used for the gaussian pdf, here you need capital phi Φ
   **Response**: thankyou for the comment. Will be revised to Φ.

Section 2.2. Operationally, how fast is this optimization approach? When you mention that the occurrence criterion in the hardest to converge does this imply that in many cases it does not converge at all?
**Response**: Convergence performance is a topic which perhaps could be discussed in the paper at greater depth. The performance of the convergence can be summarised across all catchments by Figure 8, as these three bi-variate spatial rainfall criteria comprise the objective function of the optimisation approach. It is clear however that smaller networks will converge better than larger networks, as each random swap needs to satisfy more neighbouring stations. During writing of the paper I did consider showing plots showing directly convergence performance (by say showing end objective function values plotted against catchment size), but in the end I thought the inclusion of Figure 8 conveys enough information about the spatial performance of the model and the exact mechanics or the convergence are probably of less interest to the reader.

1. I guess the branched non-sequential procedure is describe correctly, yet to be honest as a reader I got lost here and mathematically it is really not clear what exactly spatiotemporal correlation structure this grouping of primary and secondary stations produces.
   **Response**: a mathematical background for the branched approach is indeed not present. The branching was developed in order to reduce the computational complexity and expense when dealing with large station networks. Branching is an attempt to transfer information through the network as best as possible without requiring an objective function which contains all stations within the network. This reduces computational expense but also makes sense as we should place more computation effort on nearer stations than farther stations, as the spatial dependence of farther stations is less anyway. In summary, the branched method was developed over a long period of trial and error, and has little mathematical background.

2. so this approach will preserve only the lag-1 correlations?
   **Response**: no lagged correlations are preserved. The simulation of advection and storm fronts is not considered in this work, as it was not considered a priority for the end use of derived flood frequency analysis of meso-scale catchments. For smaller and urban catchments the reproduction of storm fronts is probably useful. The approach developed here however could incorporate lagged correlations within the objective function if storm advection were to be considered a priority in the models end use.
3. Just to be sure, the disaggregation you are using is not stochastic? Right? Each daily value in transformed to the hourly ones by using the deterministic functions if I understand well. If this is the case then there aren't any fluctuations. Please clarify and state that in the text.
   **Response**: I am unsure exactly to which disaggregation the reviewer is referring to here? Within the rainfall model, the only disaggregation which occurs is to generate the hyetograph of each rain event (lines 137-146, equation 7) where the event rainfall depth (WSA) is disaggregated to hourly timesteps. The hyetograph profile is deterministic (equation 17), however the wet spell peak intensity and timing are derived stochastically. If the reviewer is referring to the disaggregation of the non-rainfall climate variables, this is indeed purely deterministic.
4. So the process preserves correlations within each catchment and not in the whole network of the 699 stations, right? Up to how many stations can this method applied effectively. For example, in our latest work (Papalexiou et al., 2023) we can go up to 10,000 stations easily preserving marginals and corrections. What are exactly the theoretical components that your approach preserve?
   **Response**: Correct, the spatial consistence is preserved on a per catchment basis. In this study, the largest catchment contains 87 stations. In theory, nothing prevents the method being applied to the entire study area simultaneously. The limitation is generally the memory requirement of the computer, as the hourly time series for all stations must be loaded in memory. Depending on how many years are simulated at once, the numeric matrix required will be very large. To overcome this issue, one could apply the method to shorter time series. This has not yet been attempted, but is possible. As our intended use is derived flood frequency analysis, where each catchment is anyway modelled independently, so the benefit of running the method across the entire study is diminished. It should also be noted that modelling at the hourly timestep (as opposed to daily timestep in Papalexiou et al., 2023) adds both complexity and a non-trivial increase in computational cost. The simulated annealing optimisation used is by no means fast, as millions of swaps are trailled and by swapping events and not timesteps, recalculating the objective function across all timesteps between swapped events (and for all relevant station pairs) can be slow. The three and only theoretical components that are preserved in the model extension into space are the three bi-variate rainfall criteria described by equations 9-11.

Please see also the works of (Peleg et al., 2017) and (Paschalis et al., 2013).
**Response:** I thank the reviewer for these interesting papers regarding high resolution rainfall models. The introduction will be updated accordingly.

1. This means that monthly variations within this summer and winter period is smoothed out? If you assess the simulation monthly within this period will it match the observed monthly characteristics?
   **Response**: correct, all months of summer show the same behaviour, as do all months

of winter. A further conditioning of the model by calendar month is of course possible, however there is often the problem of a lack of observations for model fitting, as hourly observations are generally not widely available across Germany until around 2006 onwards.

Section 4.3. Can you show a graph of a synthetic time series and an observed, and maybe for a station the probabilities of the length of wet and dry spells vs the observed ones?
**Response**: I think here the reviewer is referring to section 4.2 (rainfall) and not 4.3 (non-rainfall climate variables)? As mentioned above, the internal structure of the rainfall model is not intended to exactly mimic the observed behaviour of rainfall events. As such a plot showing synthetic vs. observed time series may be counter productive. However a plot showing probabilities of wet and dry spells is a good idea and can be included in the final manuscript.

Section 5.2.

As I mentioned I feel that this is better described as a multisite model. Does this approach have any control over advection (linear or described by generic velocity fields) or anisotropy that characterize fine scale precipitation (please see Papalexiou et al., 2021). These are important points that need to be clear discussed for precipitation even as limitations of this approach.

How about the lagged correlations of precipitation?
**Response:** the performance regarding reproduction of lagged correlations of precipitation is generally poor (see plot below), especially for lag-1, which is not surprising considering that it is not considered within the objective function of the simulated annealing procedure. As advective properties are not a focus of the study (see responses above), discussion of lagged correlation has been omitted, but can be included if deemed necessary.

[Figure]

[Figure]

Section 5.3. I wonder again if the grouping in winter and summer, e.g., Fig 14 is too coarse, especially for the variables such as temperature where there typically strong monthly variations.

**Response**: This grouping in summer and winter only applies to rainfall. The selection window *w* in the k-NN resampling process is what enforces seasonality for the non-rainfall climate variables. Figure 14 is showing summer and winter seasons only in order to reduce the amount of information shown. This figure could be expanded to show the four regular seasons if requested.

Overall, this is an interesting and useful paper that improves and extends the authors previous works and has its place in the literature. There are many methodological choices that can be better justified, several points that need clarifications, some algorithmic descriptions were hard to follow, the assessment of the generated time series can be improved, and finally, I felt that it is was not clear what theoretical properties this approach exactly reproduces and what are the limitations. I believe also a discussion section will benefit the paper were the authors should summarize limitations, maybe future extension, and put their work in context with other works. My comments are optional, and the authors can ignore them, yet I deem that this work needs amendments to became clearer and more accessible.

**Response**: I again would like to thank the reviewer for his detailed and constructive comments and I believe most points made are valid. As the model presented lacks advection or direction properties, this detail should definitely be discussed to make clear the limitations of the model but also to discuss possible future model revisions. The theoretical properties reproduced will also be outlined better to the reader. I would also agree that the algorithmic descriptions needs to be improved, which will be a focus of the final manuscript revision.

Sincerely,

Simon Michael Papalexiou

**References used within responses:**

Callau Poduje, A. C.; Haberlandt, U. (2017): Short time step continuous rainfall modeling and simulation of extreme events. In *Journal of Hydrology* 552, pp. 182–197. DOI: 10.1016/j.jhydrol.2017.06.036.

Haberlandt, U.; Ebner von Eschenbach, A.-D.; Buchwald, I. (2008): A space-time hybrid hourly rainfall model for derived flood frequency analysis. In *Hydrol. Earth Syst. Sci.* 12 (6), pp. 1353–1367. DOI: 10.5194/hess-12-1353-2008.

---

## Author Response (AR1)

The authors thank both reviewers for their constructive feedback and notes. Responses to each point are provided in red below. Many of the comments led to direct changes in the manuscript and in particular the spatial consistency optimisation approach was reworked to simplify the text and improve its readability.

**Response to reviewer #1 (Simon Michael Papalexiou)**

This is a useful paper and deserves consideration. I will avoid a generic summary and list right away several comments that hopefully the authors might find useful.

**Response**: I thank the reviewer for their many constructive and insightful comments. I will respond here to each point individually. I believe the points raised by the reviewer have led to a significant improvement to parts of the paper and an overall

As a general comment, the methods described suit more multisite methods, especially for precipitation. At the hourly or finer scales, space-time precipitation has components or advection and anisotropy, and it is not clear if what these methods preserve in this regard (I wil get back to that).

**Response**: It is true that the model presented does not preserve any attributes regarding advection or anisotropy. The focus of the research was predominantly developing methods that allow for the synthesis of large stations networks (station networks up to 100), as previous versions of the single site alternating model could only achieve this to a limited extent (station networks less than 10 – see Haberlandt et al. 2008). Anisotropy could be considered in future revisions by using directly observed values for the target values of eq. 9-11 instead of using regression models, or incorporating direction into the regression models. Advection may also be considered by considering temporally lagged values of one or more of the three bi-variate rainfall dependence criteria. This topic will be discussed further in responses below.

1. please see also Papalexiou (2022) which is dedicated only to precipitation.
   **Response**: Thank you for the reference. The introduction has been updated accordingly.
2. Rainfall is described by the single-model as an alternating sequence of independent wet and dry spells. How valid this assumption is?
   **Response**: Observed auto-correlation of the event variables is very low. For instance, the median lag-1 auto-correlation across all 699 stations is 0.0269 for wet spell amount, 0.0822 for wet spell duration and 0.0482 for dry spell duration. This is now mentioned in the Methodology as the reason for not considering auto-correlation of event variables.
3. I can understand the 1 mm limit but based on what rationale the DSDmin = 4 hr was set?
   **Response**: Choosing 4 hours for the DSDmin value was a somewhat arbitrary decision. The previous study by Callau et al (2017) used a value of 1 hour. The German guideline for calculating return periods of extreme rainfall events (DWA-A 531) requires a dry period of at least 4 hours between events to be considered independent. The text has been updated to reflect the somewhat arbitrary nature of this choice.
4. if u,v are in [0,1] then where are $F_U(u)$ and $F_V(v)$. Check the notation please.
   **Response**: Thankyou for the comment. Notation has been revised.

5. What is the justification of this choice? Have you tested e.g., the Gumbel dependence and was not suitable? Did you observe asymmetries? Please justify.
   **Response**: I assume the reviewer here is referring to the choice of copula? The main driver in the choice of Khoudraji's device, is the presence of asymmetries in the observations (events with large WSA, but small WSD). As to the choice of the Gumbel copula for $C_2$, many copula families were tested by trial and error, and Gumbel was found to lead to the best performance regarding wet spell intensities. A small description has been added to the paper regarding Copula choice.
6. Similarly, what led you to the Weibull choice for DSD and WSA. Great that you've drop the 4-parameter Kappa but why Weibull is a good model for the WSA. If you consider that, e.g., the hourly wet value distribution is a specific distribution then then WSA would be its convolution. Specifically, for the Weibull there were some attempts to justify it theoretically as a rainfall distribution by Wilson & Toumi (2005); it was also used in meta statistical approaches for daily rainfall e.g., (Marani & Ignaccolo, 2015; Marra et al., 2018, 2023) and it seems it does a good job in describing the extremes but if it is suits well for the WSA it will be nice to show some evidence. Also, here you're using the 3-par version which also can end up with ζ quite larger than the min so you might have inconsistencies in low values. Can you explain please? The same point holds for the distribution choice of the WSD. Is the LN supported by the literature? By your own analysis in this dataset?
   **Response**: Similar as for the previous response, the Weibull was chosen more through trial and error than any rigorous theoretical foundation. Many distributions were explored, then goodness of fit criteria such as the Cramer-von-Mises test and visual tests such as QQ-plots were used to arrive at the choice of Weibull for the DSD and WSA and LN for the WSD. The text has been updated describing the general procedure but comparative results comparing distributions hasn't been included due to space and overall relevance.
   When the lower bound of a distributions falls below either the WSAmin or DSDmin, any simulated value below these thresholds are replaced with the threshold value, but no process exists if lower bounds are significantly higher than either the WSAmin or DSDmin.

Equation 7. So this implies that the tail of intensity is exponential? It could for this region but in general this contradicts many global studies indicating that the tails are not exponential but heavier. Exactly for this reason, in the past I explored the Generalized Gamma to allow heavier right tails and recently some Generalized Exponential distributions having similar tails (see Papalexiou, 2022). I believe the choice of model is crucial as we're risking underestimating the potential for extremes. Please explain.
**Response**: Equation 7 describes the distribution of rainfall within the event. It should be emphasised that the internal distribution of rainfall within an event is considered of lower importance due to the intended end use of derived flood frequency analysis of meso-scale catchments where an exact reproduction of the time series over small time scales isn't necessarily required. The wet spell peak of an event is modelled through a copula that describes the dependence between the ratio WSP:WSA and the wet spell duration (eqs 2, 3, 8 in the draft) and the Weibull distribution to describe the ratio WSP:WSA. Equation 7 only applies to the timesteps before and after the wet spell peak, as the wet spell peak is preserved (see figure 2 for a visual representation).

Also what is the correlation structure within the event? Clearly at hourly resolution there is strong autocorrelation within wet values (see Papalexiou, 2022).

**Response**: Auto-correlation is generally over-estimated in simulated values (see below plot of observed and simulated auto-correlation for all 699 stations up to lag 5), which is expected due to the deterministic equation used to describe the event hyetograph. I already have several ideas on how to improve the modelling of the internal structure, which hopefully would lead to improvement of the hourly timestep autocorrelation performance, and might be explored in any future publication.

[Figure]

1. lower phi is typically used for the gaussian pdf, here you need capital phi Φ
   **Response**: thankyou for the comment. Has been revised to Φ.

Section 2.2. Operationally, how fast is this optimization approach? When you mention that the occurrence criterion in the hardest to converge does this imply that in many cases it does not converge at all?
**Response**: The performance of the convergence can be summarised across all catchments by Figure 8, as these three bi-variate spatial rainfall criteria comprise the objective function of the optimisation approach. It is clear however that smaller networks will converge better than larger networks, as each random swap needs to satisfy more neighbouring stations. During writing of the paper I did consider showing plots showing directly convergence performance (by say showing end objective function values plotted against catchment size), but in the end I thought the inclusion of Figure 8 conveys enough information about the spatial performance of the model and the exact mechanics or the convergence are probably of less interest to the reader. The optimisation process is definitely not fast, and for some catchments the computation time required is rather days than hours. All catchments were able to converge at least moderately well, and none completely failed. And even though occurrence is hardest to converge, computationally it is the easiest of the three.

1. I guess the branched non-sequential procedure is describe correctly, yet to be honest as a reader I got lost here and mathematically it is really not clear what exactly spatiotemporal correlation structure this grouping of primary and secondary stations produces.
   **Response**: I thank for the reviewer for the comment. The algorithmic description has been overhauled.

2. so this approach will preserve only the lag-1 correlations?
   **Response**: no lagged correlations are preserved. The simulation of advection and storm fronts is not considered in this work, as it was not considered a priority for the end use of derived flood frequency analysis of meso-scale catchments. For smaller and urban catchments the reproduction of storm fronts is probably useful. The approach developed here however could incorporate lagged correlations within the objective function if storm advection were to be considered a priority in the models end use. Text has been updated to make clear that lagged correlations are not considered.

3. Just to be sure, the disaggregation you are using is not stochastic? Right? Each daily value in transformed to the hourly ones by using the deterministic functions if I understand well. If this is the case then there aren't any fluctuations. Please clarify and state that in the text.
   **Response**: I assume the reviewer is referring here to the disaggregation of the non-rainfall climate variables from daily to hourly, which is indeed purely deterministic. The text has been modified in several places to state this more clearly.

4. So the process preserves correlations within each catchment and not in the whole network of the 699 stations, right? Up to how many stations can this method applied effectively. For example, in our latest work (Papalexiou et al., 2023) we can go up to 10,000 stations easily preserving marginals and corrections. What are exactly the theoretical components that your approach preserve?
   **Response**: Correct, the spatial consistence is preserved on a per catchment basis. In this study, the largest catchment contains 87 stations. In theory, nothing prevents the method being applied to the entire study area simultaneously. The limitation is generally the memory requirement of the computer, as the hourly time series for all stations must be loaded in memory. Depending on how many years are simulated at once, the numeric matrix required will be very large. To overcome this issue, one could apply the method to shorter time series. This has not yet been attempted, but is possible. As our intended use is derived flood frequency analysis, where each catchment is anyway modelled independently, so the benefit of running the method across the entire study is diminished. It should also be noted that modelling at the hourly timestep (as opposed to daily timestep in Papalexiou et al., 2023) adds both complexity and a non-trivial increase in computational cost. The simulated annealing optimisation used is by no means fast, as millions of swaps are trialled and by swapping events and not timesteps, recalculating the objective function across all timesteps between swapped events (and for all relevant station pairs) can be slow. The three and only theoretical components that are preserved in the model extension into space are the three bi-variate rainfall criteria described by equations 9-11.

Please see also the works of (Peleg et al., 2017) and (Paschalis et al., 2013).
**Response:** I thank the reviewer for these interesting papers regarding high resolution rainfall models. The introduction has been updated accordingly.

1. This means that monthly variations within this summer and winter period is smoothed out? If you assess the simulation monthly within this period will it match the observed monthly characteristics?
   **Response**: correct, all months of summer show the same behaviour, as do all months of winter. A further conditioning of the model by calendar month is of course possible, however there is often the problem of a lack of observations for model fitting, as hourly observations are generally not widely available across Germany until around 2006 onwards. A short discussion is now included in section 4 (Model setup and validation).

Section 4.3. Can you show a graph of a synthetic time series and an observed, and maybe for a station the probabilities of the length of wet and dry spells vs the observed ones?
**Response**: I assume here the reviewer is referring to section 4.2 (rainfall) and not the non-rainfall climate variables. As mentioned above, the internal structure of the rainfall model is not intended to exactly mimic the observed behaviour of rainfall events. As such a plot showing synthetic vs. observed time series would be counter productive. Figure 6 has been added to the paper to show mean station probabilities for various durations of DSD and WSD.

Section 5.2.

As I mentioned I feel that this is better described as a multisite model. Does this approach have any control over advection (linear or described by generic velocity fields) or anisotropy that characterize fine scale precipitation (please see Papalexiou et al., 2021). These are important points that need to be clear discussed for precipitation even as limitations of this approach.

How about the lagged correlations of precipitation?
**Response:** the performance regarding reproduction of lagged correlations of precipitation is generally poor (see plot below), especially for lag-1, which is not surprising considering that it is not considered within the objective function of the simulated annealing procedure. As advective properties are not a focus of the study (see responses above), analysis or discussion of lagged spatial correlations has been omitted.

[Figure]

[Figure]

Section 5.3. I wonder again if the grouping in winter and summer, e.g., Fig 14 is too coarse, especially for the variables such as temperature where there typically strong monthly variations.

**Response**: This grouping in summer and winter only applies to rainfall. The selection window *w* in the k-NN resampling process is what enforces seasonality for the non-rainfall climate variables. Figure 14 is showing summer and winter seasons only in order to reduce the amount of information shown.

Overall, this is an interesting and useful paper that improves and extends the authors previous works and has its place in the literature. There are many methodological choices that can be better justified, several points that need clarifications, some algorithmic descriptions were hard to follow, the assessment of the generated time series can be improved, and finally, I felt that it is was not clear what theoretical properties this approach exactly reproduces and what are the limitations. I believe also a discussion section will benefit the paper were the authors should summarize limitations, maybe future extension, and put their work in context with other works. My comments are optional, and the authors can ignore them, yet I deem that this work needs amendments to became clearer and more accessible.

**Response**: I again would like to thank the reviewer for his detailed and constructive comments and I believe most points made are valid. Limitations of the model have been more

clearly stated but also briefly discusses the possibility of such features in future model revisions. The theoretical properties reproduced have now been better outlined to the reader. The algorithmic description of the spatial consistency optimisation approach has been significantly improved.

Sincerely,

Simon Michael Papalexiou

**References used within responses:**

Callau Poduje, A. C.; Haberlandt, U. (2017): Short time step continuous rainfall modeling and simulation of extreme events. In *Journal of Hydrology* 552, pp. 182–197. DOI: 10.1016/j.jhydrol.2017.06.036.

Haberlandt, U.; Ebner von Eschenbach, A.-D.; Buchwald, I. (2008): A space-time hybrid hourly rainfall model for derived flood frequency analysis. In *Hydrol. Earth Syst. Sci.* 12 (6), pp. 1353–1367. DOI: 10.5194/hess-12-1353-2008.

The idea of the article is original (development of a space-time rainfall model that can accurately reproduce various rainfall statistics including extreme values) and is also very well organized.

**Response:** I thank the reviewer for their supportive feedback.

L89: Is there any mechanism in your model to consider temporal autocorrelation of WSD, WSA, DSD while generating them? (Large rainfall is quickly followed by large rainfall, and vice versa) This mechanism will enable the model to reproduce long(er)-term rainfall variability (e.g. weekly, monthly) making it more versatile (Kim et al., 2020).

**Response:** At this stage there is no mechanism to consider event variable auto-correlation, however observed event variable auto-correlation is anyway low (median lag-1 auto-correlation across all 699 stations is 0.0269 for wet spell amount, 0.0822 for wet spell duration and 0.0482 for dry spell duration), so in the authors view it makes little sense to model.

L147: Why Weibull? Generalized Pareto Distribution may be a better pdf for rainfall peak values. You may want to try the L-moment diagram method to figure out the most optimal distribution of WSP.

**Response**: We thank the reviewer for the comment and suggestion. The below diagram shows the L-moment diagram for all stations used for the study. It indicates that the Generalized Pareto Distribution may well be a good choice for the variable WSP:WSA.

[Figure]

To test its suitability, the WSP was modelled over 100 realisations of 15 years using both Weibull (as in the study) and GPA distributions (as suggested by the reviewer). Looking at the absolute relative bias, abs(mean(sim) – mean(obs) / mean(obs)), with the median taken over all realisations (below figure), a slight improvement can be observed in winter, however performance is slightly worse for summer. Other metrics (e.g. 98% percentile, standard deviation etc) align with this finding, and as such it is not considered worth the effort to change the distribution at this late stage.

[Figure]

Section 2.2. Space-time rainfall synthesis via resampling: I have an impression that the model is too much oriented toward reproducing only spatial-correlation. Do you have any algorithm to ensure space-time correlation at all gauges? The algorithm does not seem to have a capacity to simulate continuous movement of storms. In other words, do the consecutive snapshot of rainfall fields resemble with each other?

**Response:** The current model is only able to reproduce spatial consistency dependent on inter-station distance alone. Direction is not considered, nor are any temporally lagged values incorporated into equations 9-11, which could aid in modelling moving storm fronts. The focus of this paper (and where most of the effort occurred) is its multi-scale performance to model both small and large rainfall gauge networks, as previous attempts had failed to do so. The nature of the simulated annealing optimisation procedure however allows for future revisions to incorporate additional characteristics such as storm front movement etc if the intended use of the output time series warrants it. The limitations of the spatial consistency approach have now been stated.

Figure 11. The systematic underestimation may be related to the first comment of this review.

**Response:** The k-NN resampling method, unlike the space-time rainfall model, aims to reproduce daily auto-correlation of the non-rainfall climate variables. The mechanism to achieve this is via the distance metric (equation 18) used for the selection of the k nearest neighbours. There is expected to be a systematic underestimation in auto-correlation, as the best ranked candidate day is not necessarily chosen due to the discrete probability distribution used (equation 19) and the conditioning on rainfall state further restricting possible candidates.

I suggest authors to add another figure that shows a diagram showing the space-time autocorrelation between Figure 8 and Figure 9.

**Response**: As the focus of this paper is only on non-lagged spatial dependence, in the author's view it makes little sense to add a plot for lagged spatial correlations, especially as the paper already has many large plots. Please also refer to the similar comment from reviewer #1 and my response above.